

# Effect of combined application of inorganic nitrogen and phosphorus to an organic-matter poor soil on soil organic matter cycling

Faiza Anwar[1], Muhammad Sanaullah[2], Hayssam M. Ali[3],
Sabir Hussain[1], Faisal Mahmood[1], Zubda Zahid[4] and Tanvir Shahzad[1]

[1] Department of Environmental Sciences, Government College University, Faisalabad, Faisalabad, Punjab, Pakistan
[2] Institute of Soil and Environmental Sciences, University of Agriculture Faisalabad, Faisalabad, Punjab, Pakistan
[3] Department of Botany and Microbiology, College of Science, King Saud University, Riyadh, Saudi Arabia
[4] Department of Agroenvironmental Chemistry and Plant Nutrition, Faculty of Agrobiology, Food, and Natural Resources, Czech University of Life Sciences Prague, Prague, Czech Republic

Corresponding author
Tanvir Shahzad,
tanvirshahzad@gcuf.edu.pk

## ABSTRACT

**Background:** Sequestering carbon dioxide ($CO_2$) in agricultural soils promises climate change mitigation as well as sustainable ecosystem services. In order to stabilize crop residues as soil carbon (C), addition of mineral nutrients in excess to crop needs is suggested as an inevitable practice. However, the effect of two macronutrients *i.e.*, nitrogen (N) & phosphorus (P), on C cycling has been found contradictory. Mineral N usually decreases whereas mineral P increases the soil organic C (SOC) mineralization and microbial biomass. How the addition of these macronutrients in inorganic form to an organic-matter poor soil affect C cycling remains to be investigated.

**Methods:** To reconcile this contradiction, we tested the effect of mineral N (120 kg N ha$^{-1}$) and/or P (60 kg N ha$^{-1}$) in presence or absence of maize litter (1 g C kg$^{-1}$ soil) on C cycling in an organic-matter poor soil (0.87% SOC) in a laboratory incubation. Soil respiration was measured periodically during the incubation whereas various soil variables were measured at the end of the incubation.

**Results:** Contrary to literature, P addition stimulated soil C mineralization very briefly at start of incubation period and released similar total cumulative $CO_2$-C as in control soil. We attributed this to low organic C content of the soil as P addition could desorb very low amounts of labile C for microbial use. Adding N with litter built up the largest microbial biomass (144% higher) without inducing any further increase in $CO_2$-C release compared to litter only addition. However, adding P with litter did not induce any increase in microbial biomass. Co-application of inorganic N and P significantly increased C mineralization in presence (19% with respect to only litter amended) as well as absence (41% with respect to control soil) of litter. Overall, our study indicates that the combined application of inorganic N and P stabilizes added organic matter while depletes the already unamended soil.

# INTRODUCTION

The world's soils hold the largest terrestrial reserves of C, which is two times higher than that in living biomass and three times than that in the atmosphere (*Batjes, 1996*, *2016*; *Lal et al., 2021*). Moreover, it is one end of the second largest exchange of C with the atmosphere after the one between oceans and the atmosphere (*Schlesinger & Andrews, 2000*). Consequently, small changes in this stock and the rate at which it exchanges with atmosphere entails large effects for global C cycle and the attendant global warming (*Guenet et al., 2018*).

There is a dual interest in storing soil organic C (SOC) in agroecosystems. First, enhancing SOC stocks by soaking up atmospheric $CO_2$ and keeping it in soils for long term may help in mitigating climate changes (*Janzen, 2004*; *Burney, Davis & Lobell, 2010*; *Powlson, Whitmore & Goulding, 2011*; *Minasny et al., 2017*). Second, higher SOC improves soils' physical, chemical and biological properties in a way that increases the agricultural and ecosystem services on sustainable basis (*Lal, 2011*; *Bagnall et al., 2021*). Addition of crop residues like rice and wheat stubbles or maize stalks is an important strategy to enhance SOC content as billions of tones of these are produced annually (*Chenu et al., 2018*; *Goswami, Mondal & Mandi, 2020*; *Korav et al., 2022*). However, numerous studies have indicated that repeated addition of crop residues does not necessarily increase SOC content as expected (*Kirkby et al., 2014*; *Crowther et al., 2019*; *Zhang et al., 2021*). There is a mineral nutrient cost of increasing SOC content that needs to be met for storing the added residues in soils (*Kirkby et al., 2013*; *Richardson et al., 2014*). However, the interaction of two major mineral nutrients added in agroecosystems *i.e.*, N & P, in terms of C cycling is not clear.

For instance, numerous studies have shown that the mineral N addition reduces soil respiration (*Treseder, 2008*; *Janssens et al., 2010*; *Ward et al., 2017*; *Zhang et al., 2021*). This reduction in soil respiration is also linked with corresponding decrease in soil microbial biomass across biomes as shown in a meta-analysis of field studies (*Treseder, 2008*). In contradiction to mineral N effects, addition of inorganic P has been shown to increase soil respiration and microbial biomass in many studies (*Cleveland & Townsend, 2006*; *Fisk, Santangelo & Minick, 2015*; *Luo et al., 2019*; *Spohn & Schleuss, 2019*). For instance, supply of inorganic P increased soil respiration from surface layers of volcanic ash soil (*Munevar & Wollum, 1977*; *Zagal et al., 2003*). Moreover, addition of inorganic P increased soil respiration in Oxisols under the forest in Costa Rica (*Cleveland & Townsend, 2006*), in tropical forests on red soils in China (*Liu et al., 2012*) and in beech forest soils of Germany (*Spohn & Schleuss, 2019*). Most of these studies have attributed this increase in respiration and microbial biomass to the microbial alleviation of P limitation. However, desorption of soluble organic compounds by inorganic P, because of stronger sorptive capacity of the latter, makes greater amounts of substrate available for microbes thereby inducing higher respiration (*Spohn & Schleuss, 2019*). Whatever the mechanism is, the contradictory

response of soil respiration (*i.e.*, soil C cycling) to addition of inorganic forms of N and P needs further exploration.

In addition to modifying soil respiration and microbial biomass in opposite directions, the addition of inorganic N and P also differ vis-à-vis the decomposition of native SOC in response to external C inputs *i.e.*, priming of the extant SOC. Briefly, when N is added along with labile organic C, the decomposition of extant SOC is suppressed when compared to C only addition treatments (*Fontaine et al., 2011*; *Shahzad, 2012*; *Shahzad et al., 2012*; *Chen et al., 2014*; *Perveen et al., 2019a*). This indicates that the mineral N addition could help stabilize the external C inputs in the soil in addition to suppressing the extant SOC. However, addition of inorganic P along with labile C has been found to enhance mineralization of the extant SOC. For instance, *Mehnaz et al. (2019)* found that addition of inorganic P along with glucose, phenol and oxalic acid further enhanced mineralization of the extant SOC than addition of labile C treatments. Similarly, in a four-year field experiment, inorganic P addition stimulated loss of the extant SOC *via* priming effect (*Luo et al., 2019*). Given this contradiction, the application of inorganic N and P, at rates higher than are required, for the sake of soil C sequestration cannot be a given.

N and P are essential nutrients that, when applied together with litter especially in an organic matter poor soil, would provide a balanced nutrient environment for microbial communities. Increased or imbalanced nutrient availability for microbial growth and activity, desorption of organic matter from soil particles and pH alterations are immediate mechanisms known to influence SOM cycling in short-term studies. Hence, it is hypothesized that there is a significant role of nitrogen and phosphorus stoichiometry in switching between SOM mineralization and immobilization in soils naturally poor in soil organic matter. So, this study aims to explore, how do these aforementioned mechanisms differ between nitrogen and phosphorus addition in SOM poor soil? And is there any synergistic or antagonistic effect when both nutrients are applied together? Additionally, how does this system respond when fresh organic matter is supplied along with or without nutrients?

We designed this study to explore the effects of addition of mineral N, P alone or in combination, in presence or absence of maize litter in an organic matter poor, calcareous soil. We hypothesized that N addition would suppress the soil C mineralization irrespective of litter addition, whereas P addition would suppress it in absence of litter owing to limited desorption of labile C from thin SOC contents while it may stimulate it in presence of litter.

# MATERIALS AND METHODS

## Soil sampling and initial physico-chemical analyses

Soil was collected from a field located at Ayub Agriculture Research Institute, Faisalabad, Pakistan (31°23′41″N, 73°3′0″E). The field is under wheat-fallow rotation for 12 years and irrigation agriculture is practiced. The wheat productivity stands at $3.95 \pm 0.28$ tons ha$^{-1}$. This area has a subtropical monsoon climate with 200 mm yearly average rainfall and soil type in an Aridisol. The effect of precipitation on soil processes is minimum owing to

regular irrigation. Surface soil (0–15 cm) was collected randomly from five points with an auger (internal diameter 4.8 cm) and homogenized to make a composite sample. All the dead or live roots, plant debris and visible rocks were removed from the soil. After sieving with 2 mm mesh, it was stored at 4 °C in an airtight polythene bag. Soil pH was determined in 1:5 (soil: water, w/v) suspensions with a pre-calibrated pH meter (Model HANNA 210). Soil texture was determined by hydrometer method (*Bouyoucos, 1962*).

Water holding capacity (WHC) of the soil was determined by the method of *Jarrell et al. (1999)*. For this, 50 g of air-dried and sieved soil was packed in plastic columns or containers that were closed with porous fabric to allow water to move up into the soil column through capillary action. They were placed in a tray filled with distilled water until soil saturation was achieved. Wet weight of the soil column was recorded after a period of drainage of excess water under gravity. Subsequently, the soil was oven dried and dry weight was recorded. WHC was calculated as the difference between dry and wet weights and expressed in percentage of dry weight of soil.

Initial SOC was determined by Walkley Black method (*Walkley & Black, 1934*). For this, 1.0 g of fresh soil was digested with 1 N $K_2Cr_2O_7$ solution. A continuous stirring of 2 min was performed after addition of 20 mL of concentrated $H_2SO_4$. After a cooling off phase of about 30 mins, 200mL of distilled water was added to the mixture followed by addition of 10 mL of concentrated orthophosphoric acid after a gentle mixing. The remaining $K_2Cr_2O_7$ in the mixture was determined by titrating it against 0.5 M ferrous ammonium sulphate after addition of 10–15 drops of ferroin indicator. The titration was terminated when the color turned from blue to red. End point was achieved as violet blue to green. Experiment was performed in triplicates and triplicate blanks without soil were run in the same way. The SOC was estimated using the following formulae.

$$SOC\ (\%) = 1.334 \times \frac{\left[V_{blank} - V_{sample}\right] \times 0.3 \times 10}{Wt \times V_{blank}}$$

where $V_{blank}$ and $V_{sample}$ are the volume of ferrous ammonium sulphate used for blank and samples respectively, and Wt is the dry weight of the soil digested for the analysis.

After physico-chemical analyses described above it was found that the soil was a sandy clay loam and of alkaline nature (pH 8.32) with 31.7 percent water holding capacity (Table 1). Moreover, soil was found to have 8.66 g C $kg^{-1}$ soil, 600 mg N $kg^{-1}$ soil, and 7.71 mg P $kg^{-1}$ soil. The C: N of soil was 14.43 (Table 1).

## Maize litter

Fresh maize leaves were dried in an oven at 50 °C for three days. These were crushed and sieved through 1 mm mesh to be used for mixing with soils before incubation. The litter had 26.76% C and 1.25% N with a C: N of 21 (Table 1).

## Soil incubation and respiration measurement

Seventy grams of dry equivalent of soil were packed in a beaker at 60% WHC before placing it in a 1 L Mason jar. Twenty-four such microcosms were prepared and placed in
**Table 1 Physiochemical characteristics of soil & litter.**

| Parameter | Value |
| --- | --- |
| **A. Soil properties** | |
| Depth | 0–15 cm |
| Sand (%) | 53.64 |
| Silt (%) | 21.08 |
| Clay (%) | 25.28 |
| Textural class | Sandy clay loam |
| pH | 8.32 |
| SOC (g C $Kg^{-1}$ soil) | 8.66 |
| Total nitrogen (mg N $kg^{-1}$) | 600 |
| Available phosphorus (mg P $kg^{-1}$) | 7.71 |
| WHC (%) | 31.7 |
| **B. Litter properties** | |
| Carbon (%) | 26.76 |
| Nitrogen (%) | 1.25 |
| C/N ratio | 21 |

an incubator at 25 °C for a week. This period of pre-incubation was meant for stabilizing and acclimatizing microbial activity before introducing experimental treatments.

After pre-incubation, soil was amended with litter (1 g C $kg^{-1}$ soil), N (120 kg N $ha^{-1}$ in the form of $(NH_4)_2SO_4$), and P (60 kg.P $ha^{-1}$ in form of single super phosphate). This amount of mineral nutrients is recommended for wheat in the Faisalabad region from where the soil was sampled, whereas the amount of C added corresponds to the organic C entering in soil *via* below and aboveground residues at this site in one crop cycle. A completely randomized block design was used. Treatments were in triplicate and included nitrogen alone (N), phosphorus alone (P), nitrogen + phosphorus (NP), litter alone (L), nitrogen + litter (NL), phosphorus + litter (PL) and nitrogen + phosphorus + litter (NPL). Un-amended soil was used as control. While mixing the salts and/or litter in the soil, control soils were also mixed to mimic the similar disturbance to microbial activity. Amended and control soils at 60% WHC were kept in airtight 1 L mason jars along with 10 mL 1 N NaOH to capture $CO_2$-C released from soils. A glass vial filled with 10 mL distilled water was also placed in jars to keep the inside atmosphere moist. Jars with 10 mL NaOH and 10 mL distilled water only served as blanks. Sealed airtight jars were incubated at 25 °C in the dark. The NaOH traps were harvested at appropriate time intervals ensuring that $CO_2$ absorption is below 10% of its total capacity of $CO_2$ absorption at the time of each harvest. The traps were replaced at days, 2, 3, 7, 9, 13, 16, 20, 23. The total amount of $CO_2$ trapped in NaOH was determined by titrating 5 mL of it against 1 N HCl after precipitating with excess $BaCl_2$ and using phenolphthalein as indicator (*Isermeyer, 1952*; *Kaneez-e-Batool et al., 2016*). Soil moisture was adjusted at 60% WHC and fresh NaOH was placed after each respiration measure. Cumulative $CO_2$ evolution represents

the total amount of $CO_2$ respired from soil over the entire incubation period and calculated by simply adding the daily measurements of $CO_2$ evolved.

## Water extractable organic C and microbial biomass

At the end of the incubation period, soil from the incubation jars was collected and stored at 4 °C for further analyses. Water extractable organic carbon (WEOC) was determined using the method of *Ghani, Dexter & Perrott (2003)*. Briefly, WEOC was extracted by shaking 5 g soil from each replicate with 25 mL of distilled water at 150 rpm for 90 min. The mixture was then centrifuged at 10,000 × g for 5 min and supernatant was filtered through a Whatman filter # 42. The total organic C concentration in the extract was determined by following the modified Walkley-Black method (*Walkley & Black, 1934*; *Jackson, 1962*).

Microbial biomass carbon (MBC) was measured by chloroform fumigation extraction method (*Vance, Brookes & Jenkinson, 1987*). Briefly, 5 g of fresh soil from each replicate was fumigated in a desiccator under vacuum conditions for 24 h. Fumigated samples were extracted with 0.5 M $K_2SO_4$ solution in 1:5 (soil: water) ratio. Another set of non-fumigated samples each containing 5 g soil from each replicate was also extracted following the same procedure. The total organic C concentration in the extract was determined following a modified Walkley-Black method (*Walkley & Black, 1934*; *Jackson, 1962*). The difference of C concentrations between fumigated and non-fumigated samples after adjusting with the extraction factor of 0.35 was considered as microbial biomass. The microbial metabolic quotient ($qCO_2$) was determined by dividing cumulative $CO_2$ evolution by MBC (*Anderson & Domsch, 1993*). Microbial metabolic quotient is an important metric to assess how efficiently microbes are utilizing soil organic matter for their growth and energy needs. Monitoring $qCO_2$ together with nutrient cycling aids us to study and predict equilibrium between immobilization and mineralization of soil organic matter that is crucial for soil organic matter storage (*Anderson & Domsch, 1993*; *Shemawar et al., 2021*).

## Enzymatic activity in soils

Extracellular enzymes serve as a biomarker for microbial activity in soil. Studying enzyme activity allows estimation of rate and efficiency of SOM decomposition and nutrient release from SOM. Dehydrogenase activity was determined as reported by *Mina & Anita Chaudhary (2012)*. Briefly, 5 g soil from each treatment was added to triphenyl tetrazolium chloride (TTC) solution (prepared by mixing 5 g of TTC in 0.2 M tris-HCl buffer, (pH 7.4)). The mixture was incubated for 12 h at 37 °C. Immediately after incubation, two drops of concentrated $H_2SO_4$ were added to cease the reaction. This mixture was shaken at 250 rpm for 30 min after adding 5 mL of toluene. The resultant mixture was centrifuged at 4,500 rpm for 5 min for the extraction of 1,3,5-triphenylformazan (TPF). The absorbance of supernatant was measured at 492 nm with UV-vis spectrophotometer. The soil dehydrogenase activity was stated as μg TPF $g^{-1}$ 12 $h^{-1}$.

Alkaline phosphatase activity was determined spectrophotometrically (*Tabatabai & Bremner, 1969*). A total of 1 g soil was mixed with 0.25 mL of toluene, 1 mL of

p-nitrophenol phosphate solution and 4 mL MUB buffer (pH 11). The mixture was incubated for 1 h at 37 °C. One mL of $CaCl_2$ (0.5 M) and 4 mL of NaOH (0.5 M) were added to the mixture after incubation. This suspension was filtered with Whatman filter paper no. 42. Absorbance of filtrate was measured at 400 nm. The phosphatase activity was expressed as µg p-nitrophenol $g^{-1}$ $h^{-1}$.

Urease activity was estimated according to "urea remaining" method by *Tabatabai (1994)*. For this, 5 g soil was incubated with 5 mL of urea solution (10 mg urea/mL) for 5 h at 37 °C. After incubation, mixture was added with 50 mL of 2 M KC1-PMA. Suspension was shaken for 1 h and filtered with Whatman filter paper no. 42. Two mL of filtrate was added with 10 mL 2 M KC1-PMA and 30 mL coloring reagent (10 mL of 0.25% TSC in 500 mL acid reagent + 25 mL 2.5% DAM). For color development, this mixture was kept in water bath for 30 min followed by ice cold water bath for 15 min. Absorbance of this mixture was observed at 527 nm with UV-Vis spectrophotometer. Soil urease activity is expressed as µg. $g^{-1}$ soil. $h^{-1}$.

*β*-glucosidase activity was measured as described by *Hayano (1973)*. For this, 0.5 g soil was mixed with 0.1 mL of toluene. 10 mL distilled water was added after 10 min followed by addition of 1.5 mL of Mcilvain buffer and 0.6 mL of PNG. This mixture was incubated at 37 °C for 1 h after vortex for few seconds. After incubation, 8 mL of ethanol was added and swirled for few seconds. The mixture was filtered, and filtrate was added with 2 mL of tris-solution. The intensity of yellow color was measured at 400 nm on UV-Vis spectrophotometer. The p-nitrophenol content of treatments were calculated by standard calibration curve method with standards containing 0, 0.2, 10.4, 0.6, 0.8, and 1.0 $\mu$ M of p-nitrophenol.

### Statistical analysis

One-way ANOVA was carried out to compare the effects of treatments on SOC mineralization (cumulative $CO_2$ and per day $CO_2$ evolution), water-extractable organic C, microbial biomass C, metabolic quotient and enzymatic activities of dehydrogenase, urease, *β*-glucosidase, and alkaline phosphatase. In case of a significant difference, a *post-hoc* least significant difference test was applied to determine the significantly different means ($n = 3$) at 95% confidence interval. Data were assessed for normality before applying the ANOVA and log-transformed to acquire normality in case needed. All the statistical analysis was conducted using Statgraphics Plus.

## RESULTS

### Soil carbon mineralization

Carbon mineralization rate (mg $CO_2$-C $kg^{-1}$ soil $d^{-1}$) was modified by nutrient (N, P) and litter addition alone as well as in different combinations ($P < 0.05$, Fig. 1A). Moreover, it varied over the course of incubation. Similarly, the cumulative $CO_2$-C release (mg $CO_2$-C $kg^{-1}$ soil) was significantly modified by nutrients and litter addition ($P < 0.05$, Fig. 1B).

The N addition (N treatment) significantly lowered the C mineralization rate (Fig. 1A) resulting in the lowest total cumulative $CO_2$-C release by the end of the experiment (Fig. 1B). Phosphorus addition (P treatment) induced ~3 times increase in C

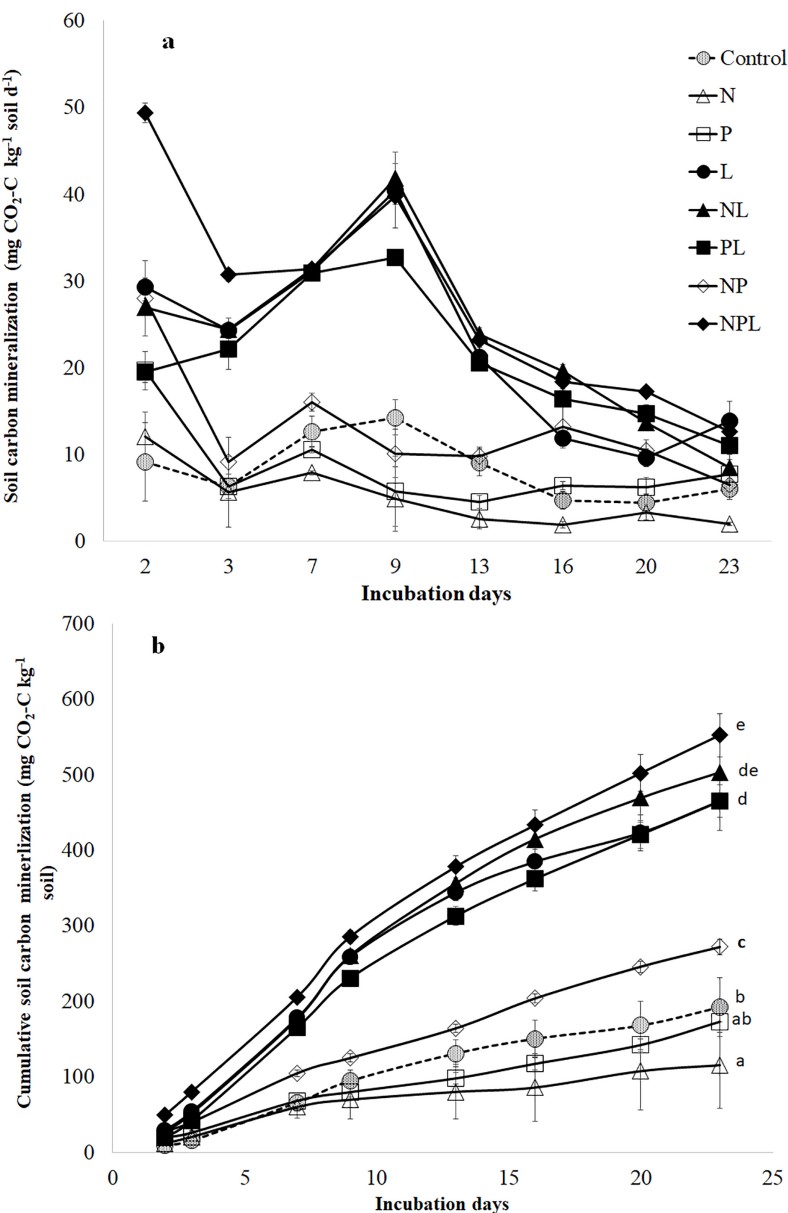

**Figure 1 Soil C mineralization in response to addition of mineral nutrients and litter addition.** Soil C mineralization rate (mg $CO_2$-C $kg^{-1}$ soil $d^{-1}$) (A) and cumulative $CO_2$-C release from control and amended soils over 23 days of incubation. Values are means of three replicates and error bars represent standard error of means ($n = 3$). The small letters in (B) show significant different among means based on one way ANOVA followed by a *post hoc* least significant difference ($P < 0.05$).

mineralization rate two days after incubation (Fig. 1A). However, afterwards, the C mineralization rate in P treatment declined to that in control treatment for the entire duration of incubation. The total cumulative $CO_2$-C released from P treatment was similar to that in control (Fig. 1B). Combined addition of N & P (NP treatment) induced higher (at times three-fold) release of $CO_2$-C in the earlier stages of incubation (Fig. 1A) leading to higher total cumulative $CO_2$-C from NP treatment than control (41%) (Fig. 1B).

The litter addition (L treatment) alone or in combination with one or both the nutrients significantly increased the C mineralization rate than control treatment (Fig. 1A). Until the 13th day of incubation, the C mineralization rate in L treatment was significantly higher than in control, N, P or NP treatments. Afterwards, the C mineralization in L treatment declined towards those in control, P and NP treatments. Overall, the total cumulative $CO_2$-C release was significantly higher in L treatment than N, P, NP and control treatments.

Adding N along with L (NL treatment) did not influence C mineralization rate when compared to litter alone (L) treatment (Fig. 1A). Moreover, total cumulative $CO_2$-C release was similar in NL and L treatments (Fig. 1B).

Although adding P along with L (PL treatment) increased C mineralization rate in the early days of incubation (Fig. 1A), the total cumulative $CO_2$-C release was similar in L and PL treatments by the end of the experiment (Fig. 1B). Adding two nutrients i.e., N & P with L (NPL treatment) significantly increased the C mineralization rate on 2nd day by 153%, 83% and 68% in comparison to PL, NL and L treatments respectively (Fig. 1A). It was still significantly higher on the third day by 39%, 26% & 26% in NPL treatment than PL, NL and L treatments respectively. Afterwards, NPL treatment showed a similar C mineralization rate as in PL, NL & L treatments. On the final day, all the litter amended soils (irrespective of the nutrient type and presence) showed similar C mineralization including NPL treatment. By the end of the experiment, the highest amount of cumulative $CO_2$-C was released from the NPL treatment, which was 19% and 187% higher than L & control treatments, respectively (Fig. 1B).

## Water extractable organic C, microbial biomass C & metabolic quotient

The treatments significantly modified the water extractable organic C (WEOC) content in soils (Fig. 2A). The N only treatment tended to lower the WEOC content although it was not significantly lower than that in control. On the contrary, addition of L with N (NL) treatment tended to increase WEOC. The three treatments i.e., P, NP, & NPL maintained the WEOC content similar to those in control. However, the PL and L treatments significantly increased the WEOC content with respect to control whereby the highest increase, which was recorded in L treatment, was 76% higher than that in the control treatment.

The microbial biomass carbon (MBC) was significantly influenced by the experimental treatments (Fig. 2B). The highest MBC was found in NL treatment that was 144% higher than control, whereas the N only treatment showed a non-significant increase of 22% only. The P only treatment significantly increased the MBC by 53% in comparison to control. However, the addition of L along with P (PL treatment) did not change the MBC. Addition of both the nutrients (i.e., NP) did not influence the MBC although it tended to increase. However, addition of L in combination with NP (i.e., NPL treatment) significantly increased the MBC by 105% with respect to control. The L treatment significantly increased the MBC by 59% in comparison to control.

Metabolic quotient ($qCO_2$) decreased significantly in response to N or P only additions ($P < 0.05$, Fig. 2C). It remained similar to control treatment in NP, L, NL and NPL treatments. However, it was significantly higher in PL treatment.

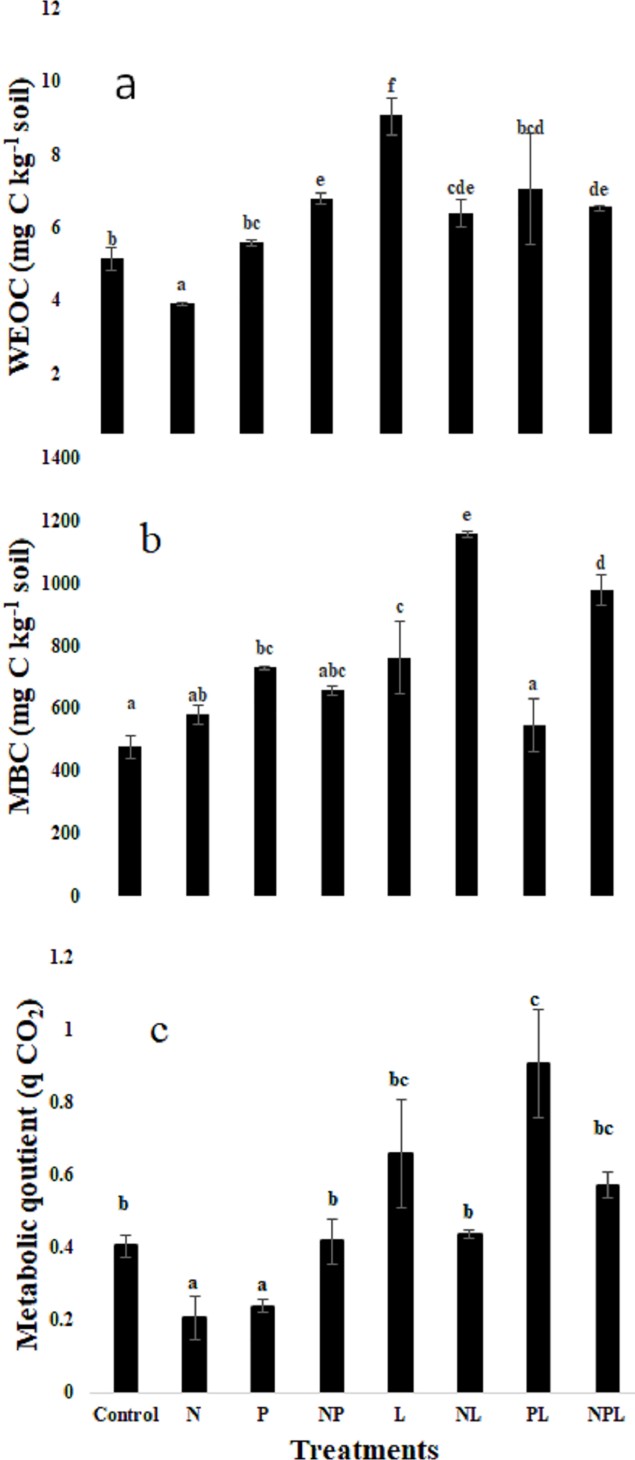

**Figure 2 Water extractable organic carbon, microbial biomass carbon and metabolic quotient in response to inorganic nutrients and litter addition.** Water extractable organic carbon (WEOC, mg C kg$^{-1}$ soil) (A) Microbial biomass (mg C kg$^{-1}$ soil) (B) Metabolic quotient (C) for control and amended soils at the end of the incubation. Letters of top of bars represent significant difference among means ($n = 3$) based on *post hoc* LSD at $P < 0.05$. Error bars show standard errors of means ($n = 3$).



### Extracellular enzymatic assays

The treatments significantly changed the activity of all the four enzymes assayed in the experiment (Fig. 3). Briefly, dehydrogenase activity significantly decreased in response to N or P and NP additions as well as when phosphorus was added with litter (PL treatment). However, it increased significantly in L, NL and NPL treatments. It was highest in NL and NPL treatments.

The N addition slightly but significantly increased the β-glucosidase activity whereas β-glucosidase activity remained unchanged in response to P or NP addition. It was 1.9, 2.1, 1.6 and 2.3 times higher in L, NL, PL and NPL treatments respectively when compared to control.

N addition significantly reduced the urease activity when added alone (N), or in combination with phosphorus and/or litter (P, L & NPL treatments) (Fig. 3C). Addition of P and L (P & L treatments) significantly increased the urease activity by 1.46 and 1.11 times respectively, with respect to control. However, the combined addition of P and L (PL treatment) did not change urease activity.

The alkaline phosphatase activity significantly decreased in response to P addition alone or in combination with N and/or L (*i.e.*, NP, PL & NPL treatments). However, adding nitrogen and litter alone or in combination significantly increased the alkaline phosphatase activity.

## DISCUSSION

### Soil carbon mineralization and C pools

Addition of inorganic P significantly stimulated soil C mineralization rate for a very brief duration followed by similar C mineralization rate as in control soils for most part of the incubation. Consequently, the total cumulative $CO_2$-C release from P treatment was similar to that in control (Fig. 1). This result is in partial contradiction to previous findings where inorganic P addition have been found to induce decrease in SOC by stimulating its mineralization (*Ehlers et al., 2010*; *Poeplau et al., 2016*; *Poeplau, Herrmann & Kätterer, 2016*; *Li et al., 2022*). According to recent research, addition of P leads to desorption of simple organic compounds making them available for microorganisms thereby leading to higher soil respiration (*Spohn & Schleuss, 2019*; *Spohn et al., 2022*; *Xia et al., 2024*). Most of the studies reporting higher C mineralization in response to addition of inorganic P used soils having high SOC content (*i.e.*, >1.5%) suggesting that high amounts of labile organic compounds were available for desorption in response to addition of inorganic P. However, our soil is poor in organic C (*i.e.*, 0.87%, Table 1) which explains the transitory stimulation in C mineralization after P addition. Later on, the labile compounds that could be desorbed from soil particles were most likely too low to sustain higher C mineralization rates. This is evident by the similar water extractable organic C (Fig. 2A). Moreover, increase in microbial biomass in P treatment indicates that the microbes used most of the desorbed organic compounds as well as the extra available P for growth (Fig. 2B). This suggests that P addition in this soil could build up microbial biomass thereby improving soil health.

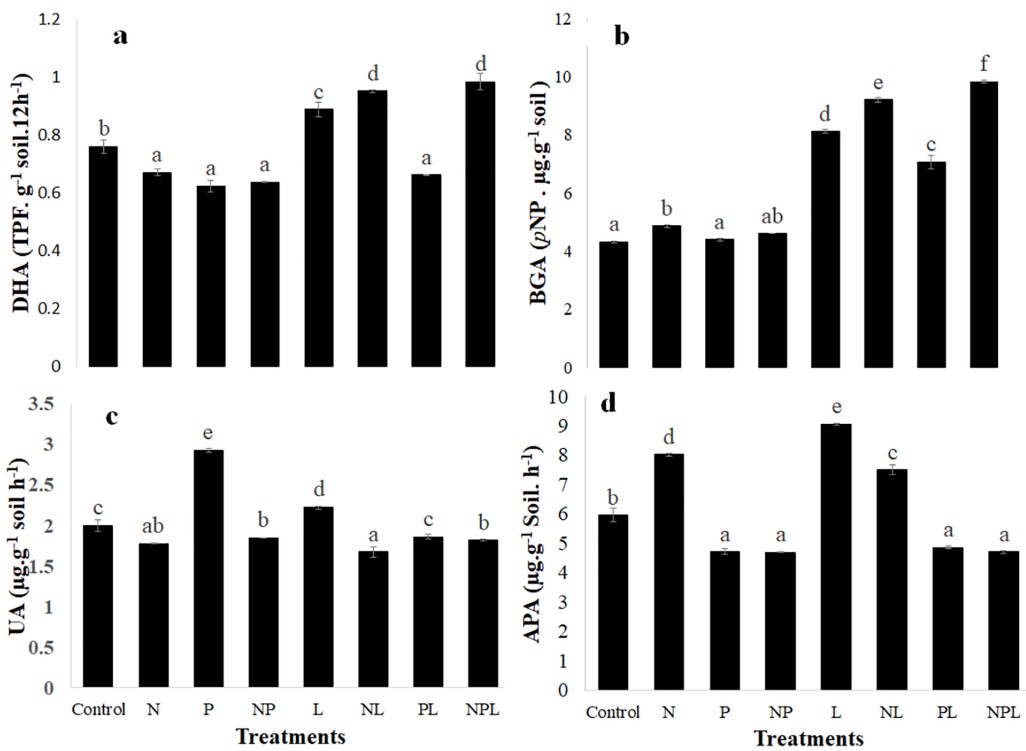

**Figure 3 Enzymatic activity in soil in response to addition of mineral nutrients litter.** Dehydrogenase (DHA) (A) β-glucosidase (BAG) (B) Urease (UA) (C) and alkaline phosphatase (APA) (D) activities in control and amended soils. Letters on top of bars indicate a significant difference among means based on *post hoc* LSD at $P < 0.05$. Error bars show standard errors of means ($n = 3$).

Mineral N addition significantly suppressed the total cumulative $CO_2$-C release as well as the soil C mineralization rate for most part of the incubation (Fig. 1). These results correspond to previous findings (*Treseder, 2008*; *Janssens et al., 2010*; *Kaneez-e-Batool et al., 2016*; *Poeplau, Herrmann & Kätterer, 2016*; *Perveen et al., 2019a*; *Wende et al., 2020*; *Fernández-Alonso, Díaz-Pinés & Rubio, 2021*; *Treseder, 2008*; *Fernández-Alonso, Díaz-Pinés & Rubio, 2021*). However, in our study, there was rather an increase, albeit an insignificant one, in microbial biomass after N addition. We assume that the soil microbes were N-limited and switched to growth instead of C mineralization as soon as the mineral N was available. However, combined addition of N & P (*i.e.*, NP treatment) induced significantly higher C mineralization whereas microbial biomass was higher (though not significantly) than control (Figs. 1, 2). Apparently, P presence in NP treatment made labile C available for microbial consumption as is shown by higher WEOC content in NP treatment than in N or P alone treatments thereby leading to higher C mineralization (Fig. 2A).

Litter addition enhanced C mineralization since it ensures provision of labile substrates for soil microbes which are C limited in general due to lower accessibility of the SOC (*Fontaine et al., 2007*; *Dungait et al., 2012*; *Shahzad et al., 2018*). Moreover, the studied soil is particularly poor in organic matter thereby increase in C mineralization in response to

litter is intuitive. However, provision of N with litter (*i.e.* NL treatment) did not cause an additional increase in C mineralization rate and cumulative $CO_2$-C release in comparison to L only treatment, although this soil is N-deficient (Fig. 1). Evidently, the microorganisms used extra available mineral N and labile C in NL treatment to build their biomass as shown by the highest microbial biomass in the said treatment (Fig. 2B). Moreover, as shown by the unchanged metabolic quotient in NL treatment, it is confirmed that microbes were more inclined to growth than decomposition. Indeed, mineral N availability in the presence of relatively labile substrate has been found to increase microbial biomass and suppress C mineralization (*Fontaine et al., 2004*, *2011*; *Dijkstra et al., 2013*; *Shahzad et al., 2015*; *Liao, Tian & Liu, 2021*).

Addition of P with litter (*i.e.* PL treatment) initially reduced C mineralization when compared to L only treatment, which is counterintuitive (*Poeplau, Herrmann & Kätterer, 2016*; *Meyer et al., 2018*; *Spohn & Schleuss, 2019*). Interestingly, microbial biomass in PL treatment decreased significantly when compared to L only treatment. Moreover, it was also significantly lower than that in P only treatment (Fig. 2B). The question is why microbial biomass decreased in response to combined addition of P and L where litter is a rich source of labile C, when addition of alone inorganic P stimulated microbial biomass? Apparently, the acute N limitation induced by excess availability of inorganic P in presence of litter can explain this result. Because, when N was also added *i.e.*, NPL treatment, this limitation was alleviated leading to the highest C mineralization and cumulative $CO_2$-C release. These results indicate that addition of both nutrients in the recommended quantities alongside organic matter stimulates microbial activity.

Litter addition significantly increased microbial biomass (Fig. 2B). However, addition of N (*i.e.*, NL and NPL treatments) further increased this living component of SOC. This shows that microbes were facing N limitation and grew their biomass when it was available. This hypothesis is confirmed by the highest WEOC content in L treatment though the MBC in the same was significantly lower than NL & NPL treatments indicating incapacity of microbes to assimilate the extra available WEOC. Overall, the results of microbial biomass show that addition of nitrogen along with litter increases the living pool of SOC.

## Enzymatic activity

Activity of C, N and P acquiring extracellular enzymes, *i.e.*, β-glucosidase, urease, and alkaline phosphatase respectively, was determined to link the effect of addition of litter, N, and P on soil processes involving cycling of these elements. Moreover, dehydrogenase activity was measured because it is linked with the active part of the microbial biomass given that it is an intracellular enzyme (*Xiang et al., 2008*).

Litter addition significantly stimulated dehydrogenase activity, except in PL treatment, evidently because the substrate availability activates microorganisms towards decomposition and growth. However, nutrient addition suppressed dehydrogenase activity presumably due to lowering of soil pH (*Wang et al., 2014*; *Tian et al., 2020*). We did not measure the soil pH post incubation. However, the salts we used to add inorganic N and P

*i.e.*, ammonium sulphate and single super phosphate respectively, are highly acidic indicating the reduction in pH in these soils.

Labile C input, litter input in our study, stimulates microbial activity and growth leading to higher microbial C demand (*Hernández & Hobbie, 2010*; *Zhou et al., 2021*). This enhances the enzyme production for co-metabolizing the SOC thereby leading to higher C degrading enzymatic activity as indicated by high β-glucosidase activity in all litter amended soils (Fig. 3B, *Perveen et al., 2019b*; *Zhou et al., 2021*). Among all litter amended soils, PL showed the lowest β-glucosidase activity which corresponds to lower microbial biomass found in this treatment (Fig. 2B). Activity of C acquiring enzyme not only increased in the presence of litter but also in the presence of mineral N (Fig. 3). Indeed, higher N availability allows microbes to invest N into enzyme synthesis thereby mining C for microbial growth (*Allison & Vitousek, 2005*; *Sinsabaugh & Follstad Shah, 2012*; *Zheng et al., 2024*). This is also confirmed by higher microbial biomass found in N treatment (Fig. 2B). An unchanged β-glucosidase activity in P treatment further confirms that the increase in microbial biomass in this treatment is because of desorption of labile organic compounds from soil particles instead of alleviation of microbial P limitation. Overall, β-glucosidase activity confirmed that adding mineral nutrients alongside organic matter (litter) in this organic matter poor soil promotes the growth soil C pools *i.e.*, microbial biomass in addition to stimulating the microbial activity.

All the treatments where mineral N was added showed significantly lower urease activity than control (Fig. 3C). This indicates that the microbes were not investing in synthesis of enzymes aimed at N mining because they were getting mineral N from outside source (*Haase et al., 2008*; *Bhaduri et al., 2016*). Instead, they were more focused on assimilating it along with labile C to increase their biomass (Fig. 2B). In all the other treatments where N was not added, except PL, urease activity was significantly higher than control indicating that microbes were investing on N acquisition because their C: N stoichiometry was highly imbalanced in the presence of excess C. A similar activity for meeting stoichiometric needs was observed in the case of P-acquiring enzyme *i.e.* where inorganic P was added, alkaline phosphatase activity was lower and vice versa (Fig. 3C). When a nutrient is limiting while other nutrients and substrate are available, it is common for microbes to synthesize enzymes to acquire the limiting nutrient (*Allison & Vitousek, 2005*; *Loeppmann et al., 2016*; *Stock et al., 2019*). This stoichiometric response by the enzymes indicates that the provided nutrient contents in the soil were sufficient to fulfil the needs of the microorganisms thereby facilitating them in their activity and growth.

## CONCLUSION

We conducted this study to find out the reason behind contradictory response of soil C mineralization and microbial biomass to addition of inorganic N and P. We found that increased C mineralization in response to addition of inorganic P, often found in literature, is indeed attributed to relatively high intrinsic SOC content from which labile desorbs thereby stimulating C mineralization. However, in an organic matter poor soil, like the one used in this study, limited desorption of labile C precludes stimulated C mineralization in response to inorganic P addition. Moreover, our study confirms that the mineral N

addition could stabilize the existing as well as externally added organic carbon as shown by suppressed mineralization and increased microbial biomass. Moreover, our study reveals that co-application of inorganic N and P in SOC poor soils can lead to loss of the soil C. However, if they are coapplied alongside external organic matter, which is the case in natural as well as most managed ecosystems, they may improve the microbial activity that's vital for soil health as well as help enhance soil C.

### Funding

This work was funded by the Researchers Supporting Project number (RSP2024R123), King Saud University, Riyadh, Saudi Arabia. The funders had no role in study design, data collection and analysis, decision to publish, or preparation of the manuscript.

### Grant Disclosures

The following grant information was disclosed by the authors:
King Saud University, Riyadh, Saudi Arabia: RSP2024R123.

### Competing Interests

Tanvir Shahzad and Sabir Hussain are Academic Editors for PeerJ.

### Author Contributions

- Faiza Anwar conceived and designed the experiments, performed the experiments, prepared figures and/or tables, authored or reviewed drafts of the article, and approved the final draft.
- Muhammad Sanaullah performed the experiments, authored or reviewed drafts of the article, and approved the final draft.
- Hayssam M. Ali conceived and designed the experiments, analyzed the data, authored or reviewed drafts of the article, and approved the final draft.
- Sabir Hussain performed the experiments, authored or reviewed drafts of the article, and approved the final draft.
- Faisal Mahmood analyzed the data, prepared figures and/or tables, authored or reviewed drafts of the article, and approved the final draft.
- Zubda Zahid performed the experiments, prepared figures and/or tables, authored or reviewed drafts of the article, and approved the final draft.
- Tanvir Shahzad conceived and designed the experiments, analyzed the data, authored or reviewed drafts of the article, and approved the final draft.

### Data Availability

The raw data are available in the Supplemental File.

## Supplemental Information

Supplemental information for this article can be found online at http://dx.doi.org/10.7717/peerj.17984#supplemental-information.

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
