# Peer review of "Effect of combined application of inorganic nitrogen and phosphorus to an organic-matter poor soil on soil organic matter cycling"

_PeerJ, doi:10.7717/peerj.17984_

## Round 0.1 · original submission · Minor Revisions

All three reviewers recommended a minor revision and provided constructive suggestions, and I anticipate that the manuscript will be improved by a thoughtful revision following the reviewer suggestions. Perhaps most important to address are the similar concerns about the lack of specific research questions and/or hypotheses in the Introduction. Please clearly identify one or more research questions and specify how they are related to, and/or motivated by, one or more knowledge gaps that are also discussed in the Introduction. If you had expectations for what the results would be as related to those questions, then also share prediction(s) and/or hypotheses. Another important focus of your revision should be on providing more information throughout the manuscript to help the reader to understand how the biological and physical properties of the focal site and soil, and how the management practices of the focal site, are relevant: 1) when describing knowledge gaps in the Introduction, 2) when describing and justifying specific components of the experimental design (e.g., the amounts of C and nutrients added to the incubations), and 3) when interpreting the results. In other words, more context based on the soil and site properties is needed, as pointed out by reviewers 1 and 3. All three reviewers also noted improvements that could be made in the quality of the figures, and suggested that the Discussion would benefit from some re-organization or other minor editing for clarity and effectiveness.

I expect that the authors may choose to provide a brief rationale for not revising the manuscript according to the reviewer suggestions in some instances, especially for reviewer suggestions that are more subjective or isolated.

**Language Note:** The review process has identified that the English language must be improved. PeerJ can provide language editing services - please contact us at [email protected] for pricing (be sure to provide your manuscript number and title). Alternatively, you should make your own arrangements to improve the language quality and provide details in your response letter. – PeerJ Staff

Reviewer 1 ·

Basic reporting

The title of the manuscript reflects only one of the experiment's outcomes and does not adequately represent the entire experimental scope. Therefore, presenting this aspect of the results as the title should be reconsidered.

Overall, the work lacks a clear working hypothesis. What do the authors expect from nutrient addition experiments specific to these soils, compared to previous studies? What outcomes are anticipated from measuring soil carbon and nutrient mineralization, as well as the microbial parameters resulting from these nutrient manipulations?

The experiment is conducted in arid soil, presenting distinct differences compared to, for example, temperate forest soils often referenced in previous studies. Adding organic amendments with low C:N ratios to this soil raises questions about the expected outcomes. Given the arid nature of the soil, how might its characteristics influence the response to such amendments, particularly in terms of nutrient cycling and microbial activity?

The annual precipitation of 200mm is exceptionally low for agricultural soil. However, considering that the soils were artificially watered during the experiment, does this climatic information still hold relevance, or does the artificial watering negate its significance?

Experimental design

The experimental design is deemed adequate. To enhance clarity, brief descriptions (one sentence each) of more cryptic or specific parameters, such as microbial metabolic quotient and extracellular enzyme analysis, are suggested to be included in the methods section to aid general readers' understanding.

Validity of the findings

Line 198: “The nutrient (N, P) and litter addition as well as the time of incubation significantly changed the C mineralization rate (mg CO2-C kg-1 soil d-1)..” time is not the experimental treatment and therefore does not change the C mineralization rate.

The authors made two assumptions: firstly, on line 346, that nutrient addition suppressed dehydrogenase activity possibly by lowering the pH associated with nitrogen and/or phosphorus addition. The change in pH following nutrient addition could be easily measured to confirm this assumption. Secondly, on line 376, they assumed that soil organic carbon (SOC) richness determines the response of its mineralization to inorganic phosphorus addition. However, the significance of SOC in soil was not empirically tested. These assumptions (including the change in soil pH) could be validated through additional data analysis, such as redundancy analysis (RDA) or even a simple regression analysis.

Additional comments

The conclusion consists repeated summary of the main findings and lacks the following points: why was the research done, and what implications of the findings concerning previous studies and particularly soil carbon sequestration in this specific region.

Please enhance figure quality. The standard error in Fig. 2 is evident and appears retouched. Increase the size of x and y variables in all plots, along with axis titles and legends. Ensure abbreviation of legends is included in the figure caption. Also, utilize distinct color gradients or textures for bars in both Fig. 2 and Fig. 3.

Reviewer 2 ·

Basic reporting

o English language may be improved.
o Introduction and background are enough. The author may improve this section by including a knowledge gap statement.
o Include some recent reference as some references are more than 3 decades old. This would improve thesis quality.
o Figures might be improved to ensure manuscript quality.
o A sketch demonstrating study mechanism would improve manuscript quality.

Experimental design

o The authors might consider providing more details about study design. Authors might include experimental design, replication details in Materials and Method’s section.
o Identify the knowledge gaps, followed by research question and scientific hypothesis.
o Include more details about adapted methods for water holding capacity and SOC was measured.

Validity of the findings

o Biological replications count is enough to validate an experiment. Authors might enrich manuscript by include technical replications in Materials and Methods’ section.
o Conclusions are well stated, linked to original research question and limited to support results.
o Authors might consider providing more details of statistical methods such as if data set normality was assessed, data outliers were identified, which post-hoc test was used to compare treatment means, how many technical replications were used in laboratory analyses.
o Conclusion are well stated and linked to research topic.

Additional comments

o Would be better to change title to more specific as “Effects of nitrogen and phosphorus applications on carbon mineralization in an organic matter poor soil”.
o Authors need consistency in abbreviation use. For instance, define abbreviation at first use and only use abbreviation onwards.
o L25: Define carbon dioxide “Sequestering carbo dioxide (CO2)…..”.
o L26: “…………… soil carbon (C)……..”
o L28: “ i.e., nitrogen (N) and phosphorus (P)……………….”.
o Author included background knowledge; however, missed the research question(s) / objective(s) in L30. The author might consider including research question(s)/objective(s) to improve manuscript quality.
o L32: soil organic carbon (SOC)……………?
o L33: in laboratory incubation experiment?
o L34: “. ………………. And incubated for ??? days and soil respiration was recorded”?
o L35: Contrary to literature………………. Is this literature related to soils poor in soil organic carbon?
o L43: Co-application of inorganic N and P?
o L41: “……… increased C mineralization……….”?
o L43: …..”(19% with respect to only litter amended)………”?
o L34 to L: Use only one term to improve manuscript quality “un-amended or control, un-amended control soil”.
o L35 – L37 and L42-L43: The author needs to rephrase sentences for more clarity i.e., either briefly increased or does not increase to avoid contradictions.
o L48: “.. reserves of carbon (C)……”?
o L49: Author might consider using latest reference instead Batjes 1996. There are several new studies which can be cited.
o L59: Authors might consider stating maize residues as current study used maize leaves?
o L65: “nitrogen (N) and phosphorus (P)….”
o L67-L68: Author might include some more references when referring to “…. numerous studies………..”?
o L67 – L70: Author might consider avoiding a 2-sentence paragraph and place it with some other related paragraph.
o L73 – L75: The author might consider using a latest reference. Current reference Munevar & Wollum (1977) is 46 years old.
o L85: “….. soil organic matter (SOM)………”?
o L98: “… explore the effects…”?
o L99 – L102: Seems like text related to Material and Method’s section?
o L98 – L102: The author should include specific research question and scientific hypothesis in this section.
o L104 -L 118: Author should include text related to soil initial properties. What was initial P and N concentrations, what were properties of maize litter used in this study.
o L104 – L118: Authors should include a shot paragraph here including information on which experiment design was used in this study, how many biological or technical replications were used to capture variability in experiment and laboratory analyses, if jars were repositions to avoid position effects, where experiment was conducted i.e., university/research institute, either experiment was randomized? All this information would help to replicate the experiment.
o L112: Include pH meter model and maker.
o L113 – L115: Include a brief method how water holding capacity and SOC was measured.
o L121: Authors might include rationale of selecting litter rate of 1 g C Kg-1 soil.
o L121: Authors might consider including information on if N and P was applied at recommended rate of what?
o L124-L125: “Un-amended soil was used as control”, candidate might consider “Un-amended soil was used as control and indicated as control onwards in the text”. Use only control in text to avoid confusion for readers.
o Use correct SI unit abbreviations i.e., mL instead of ml.
o L126-L127: 10 mL water was placed in jar in what? Small beaker or was added in the jar itself?
o L131: How cumulative CO2-C release was measured or calculated? Authors might consider including method in Materials and Method’s section.
o L136: “…….. of the incubation period,…”? Soil was harvested or collected?
o L139: mL versus ml and min versus minutes and h versus hours in L 145.
o L144: Briefly, 5 g …………..?
o L154: Briefly, 5 g …………..?
o L157: h versus hours?
o L185 – L189: “……. compare the effects…………..”; “…………. On SOM mineralization …………..”.
o L185 – L189: Authors might improve this section including information on how many biological and technical replications were carried out. If data set was analyzed for normality, no more than 5% outliers, prior to run ANOVA. What type of post-hoc text was used to compare treatment means?
o L192 – L 196: Text seems more related to Material and Method’s section. Information related to initial stock is not results rather material and methods used in this study.
o L198 - L201: Authors might consider rephrasing sentence for more clarity. For instance, experimental treatments showed significant effects on ……………………? Remember, including “nutrient (N, P) and litter addition”, do not truly represent the complete treatment list “with or without litter, control, without N, without P, etc.”. This reviewer also wonders if authors included incubation time as a treatment. If not, authors should avoid stating “as well as the time of incubation significantly…”? In case authors decide to analyze incubation over the time, it should be a separate figure showing lettering over the different sampling dates.
o L258 and L261: The N addition……………..?
o L291 – L292: These results correspond to previous findings where researchers found ………………………………………………… (References).” Authors should populate the text to show these findings corresponding to previous findings.
o L304: Authors need to rephrase sentence to avoid confusion “Litter addition, across all combinations.? Litter was not included in all treatments.
o L382: N and P?
o Figure 1: Authors might consider improving figure title i.e., “ . presence of litter and not presence of litter addition. Include which post-hoc test was used to compare treatment means. Error bars represent individual error bar for each 3 replicates. Improve X-axis as “Days of incubation”? Figure 1a and Figure 1b can be replaced on top of each other to make lettering visible as in current form, it is difficult to read letters at 150% zooming panel. Use complete name for consistency “soil carbon” as did in Figure 2.
o Figure 2: Same comments as Figure 1. Write X-axis and also include X-axis line in a and b panel.
o Figure 3: Same comments as Figure 1. Write X-axis and also include X-axis line in all panels.
o Table 1: Discuss this table in Material and Methods and Discussion section.

Reviewer 3 ·

Basic reporting

This study looked at the effects of N, P, and organic C additions on C mineralization in a soil incubation experiment. Overall, the experiment seems sound. The English is overall good although many times there are missing basic words like “the”, “a”, and “and” which would make the reading flow better but does not actually take away from the ability of the reader to understand the project, so that isn’t a huge concern of mine but something worth noting.

Figure Comments: In figure 1, panel b is not identified in the text. Also, for all figures I would suggest putting a comma or semi colon between each description of the different panels. On figure 2, add “and” after “(b)”. It is very hard to see the differences in significance letters in Figure 1b, I would suggest increasing the font size for these.

Experimental design

Research questions were not clearly stated. The introduction set the stage for the authors questions, but a statement of objectives hypotheses should be provided in the final paragraph of the introduction.

The experimental design is straight forward, and the methods used are standard. I did find the methods a little “choppy” to read, ie it didn’t always flow well, and it could be streamlined. For example, L120-L134 mentions that soils were incubated at 60 % WHC at least three times. So, this type of language could be condensed into one sentence. Furthermore, the first sentence states how the soils were incubated and then goes to treatments and then back to how they were incubated. The clarity could be improved by maybe describing the treatments first and then going into the method of incubation. This is not a huge concern of mine but would be helpful to the readers. Finally, for methods there is a tendency of the authors to switch between active and passive voice and I would suggest using all passive voice.

Validity of the findings

For the discussion, it was sometimes hard to follow what the overarching conclusions were from the different treatments. I wonder if it would be helpful to have subheadings in the discussion, which addressed either the metric of response to the treatments (e.g. C mineralization, microbial biomass, and enzyme activity) OR organize sections by responses to treatments (e.g. N response, P response, Litter response, combined responses). I am not sure the best way forward here, but do think there could be better organization of the discussion to more clearly be able to follow the major points the authors are making.

Additional comments

L106-107: Some more background on how this land had been managed would be helpful. For instance, number of years it has been in this type of land use and what type of management practices are being used (e.g. fertilization rates, harvest rates, etc).
L111: were roots, rocks etc. removed before or after sieving, I would think before?
L212: not sure what is meant by “till”. Until? Or up through?
L235: replace “it” with “WEOC”
L240: define MBC earlier in methods.
L259: replace “it” with “δ-glucosidase activity”
L284: change “that” to “which” or something else, right now it reads as a run on sentence.
L304: remove intuitively.
L306: the authors state the additions of C stimulate microbial activity due to the complexity of SOC. First off, this is a vague statement. Do the authors mean the molecular complexity of different SOC compounds or interactions with minerals in soil, or both? Furthermore, as the authors have stated that their soils had low SOC concentrations then to me this simply implies that the microbes are just generally C limited. That seems like a clearer explanation or at least worth stating in conjunction with the other statement as a possible explanation for the response to litter addition.
L315: just for clarity, the NL treatment in this study did not suppress cumulative C min, at least not statistically.
L346: change “lowering down” to “reduction”

---

## Round 0.2 · accepted · Accept

The revision led to a substantial improvement in the key areas highlighted by the reviewers. The manuscript reads well; the study was straightforward and its description was also straightforward.